# Efficient Logit-based Knowledge Distillation of Deep Spiking Neural Networks for Full-Range Timestep Deployment

Chengting Yu [1 2 †]  Xiaochen Zhao [2 †]  Lei Liu [2]  Shu Yang [2]  Gaoang Wang [2]  Erping Li [1 2]  Aili Wang [1 2 ✉]

## Abstract

Spiking Neural Networks (SNNs) are emerging as a brain-inspired alternative to traditional Artificial Neural Networks (ANNs), prized for their potential energy efficiency on neuromorphic hardware. Despite this, SNNs often suffer from accuracy degradation compared to ANNs and face deployment challenges due to fixed inference timesteps, which require retraining for adjustments, limiting operational flexibility. To address these issues, our work considers the spatio-temporal property inherent in SNNs, and proposes a novel distillation framework for deep SNNs that optimizes performance across full-range timesteps without specific retraining, enhancing both efficacy and deployment adaptability. We provide both theoretical analysis and empirical validations to illustrate that training guarantees the convergence of all implicit models across full-range timesteps. Experimental results on CIFAR-10, CIFAR-100, CIFAR10-DVS, and ImageNet demonstrate state-of-the-art performance among distillation-based SNNs training methods. Our code is available at https://github.com/IntelliChip-Lab/snn_temporal_decoupling_distillation.

## 1. Introduction

Spiking Neural Networks (SNNs) are modeled after biological neural systems and feature spiking neurons that replicate the dynamics of biological neurons (Maass, 1997; Roy et al., 2019). In contrast to Artificial Neural Networks (ANNs), which utilize continuous data forms, SNNs employ a spike-coding approach, using discrete binary spike trains for data transmission (Panzeri & Schultz, 2001). This binary signaling significantly reduces the multiply-accumulate op-

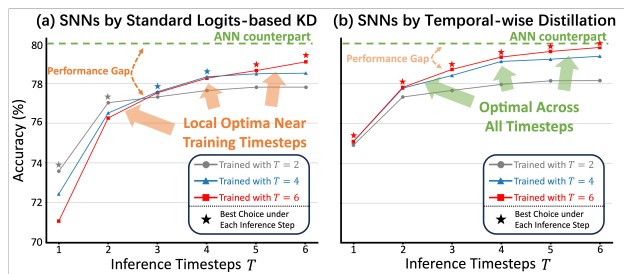

*Figure 1.* **Illustration of the primary challenges and motivations.** (a) Standard logits-based knowledge distillation (logits-KD) training suffers from large accuracy degradation and requires different models to adapt to various inference timestep settings. (b) The proposed distillation framework reduces the gap and ensures a single model for full-range timesteps.

erations generally required for synaptic connections (Roy et al., 2019), boosting both energy efficiency and speed of inference on neuromorphic hardware (Akopyan et al., 2015; Davies et al., 2018; Pei et al., 2019). In essence, SNNs can be regarded as a type of quantized model that uses binary transmission, offering the potential for low power consumption and reduced latency when implemented on neuromorphic devices (Akopyan et al., 2015; Davies et al., 2018; Pei et al., 2019).

Although SNNs exhibit considerable potential, their practical application is hindered by the non-differentiability of spike activity (Zuo et al., 2024; Deng et al., 2023), coupled with the limited expressiveness of binary spike feature maps (Qiu et al., 2024a). Together, these challenges result in accuracy degradation when compared to full-precision ANNs (Xu et al., 2023b; Zuo et al., 2024; Deng et al., 2023; Guo et al., 2023a; Hu et al., 2024b). Besides, when deploying SNNs on neuromorphic hardware, a non-negligible challenge is that the inference timesteps of the models are fixed, aligning with those utilized during training to optimize performance. Altering inference timesteps based on specific needs requires retraining the models for new timesteps (Fig. 1a), which restricts deployment flexibility and affects operational adaptability in practical settings.

As the common strategy for model lightweighting, Knowledge Distillation (KD) (Hinton, 2015; Gou et al., 2021) has been increasingly applied to training SNNs (Xu et al., 2023b; Hong et al., 2023; Guo et al., 2023a; Qiu et al.,

---

† Equal contribution. [1]College of Information Science and Electronic Engineering, Zhejiang University [2]ZJU-UIUC Institute, Zhejiang University. Correspondence to: Aili Wang <ailiwang@intl.zju.edu.cn>.

*Proceedings of the 42nd International Conference on Machine Learning*, Vancouver, Canada. PMLR 267, 2025. Copyright 2025 by the author(s).

2024a). The KD-based SNNs training leverages rich information from an ANN teacher to train a student SNN, achieving promising results on benchmark vision datasets such as CIFAR10/100 and ImageNet with CNN-based models (Xu et al., 2023b; Hong et al., 2023; Deng et al., 2023; Guo et al., 2023a; Xu et al., 2024). However, current SNNs distillation methods primarily adopt strategies from ANNs, sticking to an end-to-end framework that utilizes the SNN's ensemble outputs or averaged feature maps as distillation targets (Xu et al., 2023a; Zuo et al., 2024; Xu et al., 2023b; Hong et al., 2023; Qiu et al., 2024a). Given the unique spatio-temporal characteristics of SNNs, there is a pressing need to develop distillation approaches that more effectively leverage the distinct properties. Recent studies have demonstrated improved model convergence by isolating the truth label objectives to operate independently at each timestep (Deng et al., 2022; Xiao et al., 2022; Meng et al., 2023; Zhu et al., 2023). Drawing inspiration from these techniques that utilize temporally decoupled objectives, we recognize that fully leveraging the spatio-temporal characteristics inherent in SNNs through decoupling overall voting objectives is crucial to unlocking the full potential of SNNs distillation. Furthermore, inspired by self-distillation approaches (Zhang et al., 2019), recent works have shown the effectiveness of using additional branches based on the SNN backbone to generate extra logits for distillation (Zuo et al., 2024; Deng et al., 2023). From the perspective of ensemble learning (Allen-Zhu & Li, 2020; Wang & Yoon, 2021; Ding et al.), we further exploit the temporal properties of SNNs by considering the final voting outputs as an integration of temporal outputs across time. We recognize that the final ensemble logits can serve as soft labels for self-distillation, acting as a regularization mechanism to guide model convergence without additional computational branches or training costs

Based on the above considerations, we devised a distillation framework emphasizing temporal-wise decoupling while incorporating three types of labels: truth target, teacher label, and ensemble label. The proposed distillation framework segments the overarching training objectives into timestep-specific targets, thereby promoting uniform model performance across all timesteps and alleviating the constraints of fixed timesteps during deployment. For instance, a model trained at $T = 6$ can simultaneously produce models for $T = 2$ and $T = 4$ with accuracies that rival those explicitly trained for each timestep (see Fig. 1b). To sum up, we provide an efficient distillation framework to tackle the deployment challenges of SNNs, which not only reduces the performance gap between ANNs and SNNs but also ensures that the internal full-range timestep models within the SNN are well-trained, allowing for flexible adjustment of inference timesteps upon deployment according to specific requirements. Our contributions can be summarized as follows:

- We propose a distillation framework that emphasizes temporal-wise decoupling of objectives, which utilizes the spatio-temporal properties of SNNs and ensures implicit full-range performance without the need for retraining for specific timesteps.

- We analyze the convergence of the proposed method to show the superior efficiency and potential for better generalization. Both theoretical proofs and empirical validations illustrate the training guarantees the convergence of all implicit models across full-range timesteps.

- We conduct experiments on CIFAR-10, CIFAR-100, CIFAR10-DVS, and ImageNet, achieving state-of-the-art results among distillation-based SNNs training methods.

## 2. Related Work

**Learning Methods for SNNs.** SNNs are typically trained using two main approaches: (1) conversion methods that create a link between SNNs and ANNs through defined closed-form mappings, and (2) direct training from scratch employing Backpropagation Through Time (BPTT). Conversion methods develop precise mathematical formulations for spike representations (Lee et al., 2016; Thiele et al., 2019; Wu et al., 2021a; Zhou et al., 2021; Wu et al., 2021b; Meng et al., 2022), which enable a smooth transition from pre-trained ANNs to SNNs and support comparable performance on extensive datasets (Cao et al., 2015; Diehl et al., 2015; Han et al., 2020; Sengupta et al., 2019; Rueckauer et al., 2017; Deng & Gu, 2021; Li et al., 2021a; Ding et al., 2021). However, the accuracy of these mappings is not always guaranteed under conditions of ultra-low latency, often requiring longer durations to collect sufficient spikes and potentially reducing performance (Bu et al., 2023; Li et al., 2022; Hao et al., 2023b;a; Jiang et al., 2023). Direct training methods, on the other hand, enable SNNs to achieve robust performance with very few timesteps by using BPTT in conjunction with surrogate gradients to compute derivatives for discrete spiking events (Neftci et al., 2019; Shrestha & Orchard, 2018; Wu et al., 2018; Gu et al., 2019; Yin et al., 2020; Zheng et al., 2021; Zenke & Vogels, 2021; Li et al., 2021b; Suetake et al., 2023; Wang et al., 2023b; Zhang & Li, 2020; Yang et al., 2021). This strategy allows for the development of SNN-specific components, such as optimized neurons, synapses, and network architectures, which improve performance (Guo et al., 2023a; Fang et al., 2021b;a; Duan et al., 2022; Yao et al., 2021; Yu et al., 2022; Guo et al., 2022a; Yao et al., 2022; Shen et al., 2023; Qiu et al., 2024b; 2025; Yao et al., 2025). Despite the advantages of reduced latency, direct training incurs significant memory and

computational burdens due to the necessity to manage the backward computational graph (Li et al., 2021b; Kim et al., 2020; Xiao et al., 2021; 2022; Meng et al., 2022; Deng et al., 2023). To reduce the training expenses associated with direct methods, several recent studies have proposed various light training strategies that have gained significant attention (Mostafa, 2017; Rathi & Roy, 2021; Wang et al., 2022; Zenke & Ganguli, 2018; Bellec et al., 2020; Bohnstingl et al., 2022; Yin et al., 2023; Xiao et al., 2022; Meng et al., 2022; Zhu et al., 2023; Yu et al., 2024).

**Knowledge Distillation for SNNs.** Knowledge distillation (KD) is a well-established transfer learning technique effectively utilized for model compression (Hinton, 2015; Liu et al., 2019; Sun et al., 2019; Wang & Yoon, 2021; Wei et al., 2018). Recent works have adapted KD to train SNNs (Kushawaha et al., 2021; Lee et al., 2021; Takuya et al., 2021; Zhang et al., 2023; Xu et al., 2023b; Hong et al., 2023; Guo et al., 2023b; Xu et al., 2024; Yu et al., 2025), employing logits-based distillation from well-trained ANNs or compressing larger SNNs into more compact models. (Xu et al., 2023b) integrated both logits-based and feature-based knowledge distillation into SNNs. (Hong et al., 2023; Guo et al., 2023b; Xu et al., 2024) further puts forward layer-wise feature-based ANN-to-SNN distillation framework. However, SNNs' binary spike representation challenges the direct feature alignment with ANNs, making such detailed alignments potentially overly restrictive (Yang et al., 2025). This work thus focuses solely on logits-based distillation to explore its full potential. Furthermore, self-distillation strategies (Zhang et al., 2019; Allen-Zhu & Li, 2020; Wang & Yoon, 2021) that do not rely on teacher labels have been adapted for SNNs (Deng et al., 2023; Dong et al., 2024; Zuo et al., 2024; Ding et al.). (Deng et al., 2023) adds auxiliary branches to SNNs to generate projection logits for self-distillation through KL divergence. (Zuo et al., 2024) extends inference times to use longer timestep outputs as teaching signals for shorter timesteps. Nonetheless, these strategies increase the computational burden, elevating the training costs associated with SNNs. (Ding et al.) first treats SNNs at different timesteps as submodels from an ensemble perspective, proposing KL divergence between adjacent steps ($t$ and $t-1$) to improve performance. While (Ding et al.) focuses on inter-submodel relations, this work further explores the link between the overall ensemble output and each submodel, aiming to fully exploit the potential of temporal decoupling within the distillation framework.

## 3. Method

### 3.1. From Standard to Temporal-wise Distillation

**Standard Logits-based Distillation**: First, we consider the standard logits-based knowledge distillation setup for spiking neural networks. Given the output of the SNNs at each

timestep, $\mathbf{z}^S(t)$, and the output logits of the teacher ANNs, $\mathbf{z}^A$, the logits-based distillation loss is composed of hard and soft label components defined on the ensemble voting outputs $\mathbf{z}_{\text{ens}}^S = \frac{1}{T} \sum_t \mathbf{z}^S(t)$. The hard label corresponds to the cross-entropy loss with softmax $\mathbf{S}(\cdot)$ applied to the classification task with the true ground one-hot label $\mathbf{y}$:

$$\mathcal{L}_{\text{SCE}} = \mathcal{L}_{\text{CE}}\left(\mathbf{S}(\mathbf{z}_{\text{ens}}^S), \mathbf{y}\right) = -\sum_i y_i \log S_i(\mathbf{z}_{\text{ens}}^S) \quad (1)$$

Here, the softmax function $\mathbf{S}(\mathbf{z}) = [S_1(\mathbf{z}), \dots, S_n(\mathbf{z})]$ where $S_i(\mathbf{z}) = \frac{e^{z_i}}{\sum_j e^{z_j}}$. For the soft labels in distillation, we generally use the Kullback–Leibler divergence with a temperature scaling factor $\tau$ defined as:

$$KL\left(\mathbf{S}(\mathbf{z}_{\text{ens}}^S/\tau), \mathbf{S}(\mathbf{z}^A/\tau)\right)$$
$$= \tau^2 \sum_i S_i(\mathbf{z}^A/\tau) \log \frac{S_i(\mathbf{z}^A/\tau)}{S_i(\mathbf{z}_{\text{ens}}^S/\tau)} \quad (2)$$

Since the entropy regularization term in the KL divergence formula is only related to $\mathbf{z}^A$ and does not contribute to the SNNs training, it can be omitted. This simplifies to:

$$\mathcal{L}_{\text{SKL}} = \mathcal{L}_{\text{KL}}\left(\mathbf{S}(\mathbf{z}_{\text{ens}}^S/\tau), \mathbf{S}(\mathbf{z}^A/\tau)\right)$$
$$= -\tau^2 \sum_i S_i(\mathbf{z}^A/\tau) \log S_i(\mathbf{z}_{\text{ens}}^S/\tau) \quad (3)$$

Combining the classification and distillation losses, the total loss for SNNs standard logits-based distillation can be expressed as:

$$\mathcal{L}_{\text{SKD}} = \mathcal{L}_{\text{SCE}} + \alpha \mathcal{L}_{\text{SKL}} \quad (4)$$

where $\alpha$ is a coefficient used to balance the two losses.

**Temporal-wise Distillation** While standard logits-based distillation typically treats SNNs as purely spatial, end-to-end models, it overlooks the unique spatio-temporal characteristics inherent to SNNs. Insteads of ANNs with only spatial logits, SNNs produce multiple sets of logits over time. This could offer a unique opportunity for SNNs distillation to leverage spatio-temporal features further. Inspired by ensemble learning (Allen-Zhu & Li, 2020; Wang & Yoon, 2021), viewing the mean output of SNNs as an ensemble aggregated through voting over time, it becomes apparent that the overall outcome across these temporal dimensions tends to improve as the accuracy at each individual timestep increases. This insight allows us to intuitively redefine logits-based distillation targets to encompass outputs across various timesteps, thus transforming standard logits-based distillation into temporal-wise distillation. In this context, we define temporal-wise cross-entropy (TWCE) for hard targets as:

$$\mathcal{L}_{\text{TWCE}} = \frac{1}{T} \sum_t \mathcal{L}_{\text{CE}}\left(\mathbf{S}(\mathbf{z}^S(t)), \mathbf{y}\right) \quad (5)$$

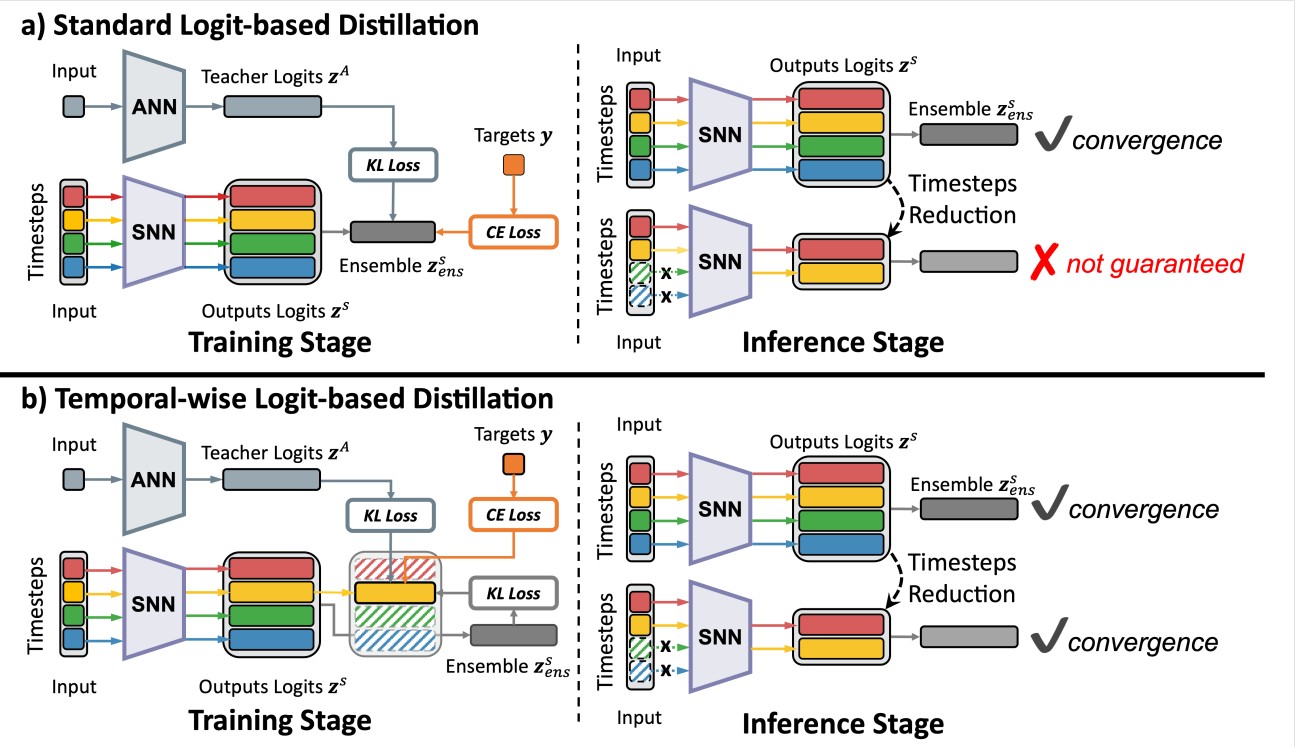

*Figure 2.* **Framework overview.** (a) Standard Logit-based Distillation defines targets on the final ensemble outputs, where model convergence is not guaranteed with reductions in inference timesteps. (b) Temporal-wise Logit-based Distillation decouples the targets into each temporal output, resulting in the guaranteed convergence of all implicit full-range timestep models.

Similarly, temporal-wise KL divergence for soft labels from an ANN teacher is formulated as:

$$\mathcal{L}_{\text{TWKL}} = \frac{1}{T} \sum_t \mathcal{L}_{\text{KL}} \left( \mathbf{S} \left( \mathbf{z}^S(t)/\tau \right), \mathbf{S} \left( \mathbf{z}^A/\tau \right) \right) \quad (6)$$

The overall objectives for temporal-wise distillations are thus combined to form:

$$\mathcal{L}_{\text{TWKD}} = \mathcal{L}_{\text{TWCE}} + \alpha \mathcal{L}_{\text{TWKL}} \quad (7)$$

### 3.2. Enhancing the Overall Framework through Self-Distillation with Final Ensemble Logits

In fact, the potential of the temporal-wise distillation framework can be further explored. Consistent with findings from student-ensemble experiments (Allen-Zhu & Li, 2020; Guo et al., 2020; Wang & Yoon, 2021), we observed that the efficacy of voting logits generally surpasses that of individual logits at separate timesteps. Consequently, we propose to further incorporate the final voting logits as an additional set of soft labels for self-distillation—beyond the true labels and teacher-generated labels—to guide the model towards improved convergence:

$$\mathcal{L}_{\text{TWSD}} = \frac{1}{T} \sum_t \mathcal{L}_{\text{KL}} \left( \mathbf{S} \left( \mathbf{z}^S(t)/\tau \right), \mathbf{S} \left( \mathbf{z}_{\text{ens}}^S/\tau \right) \right) \quad (8)$$

Accordingly, the overall training objective is formulated as:

$$\mathcal{L}_{\text{final}} = \mathcal{L}_{\text{TWCE}} + \alpha \mathcal{L}_{\text{TWKL}} + \beta \mathcal{L}_{\text{TWSD}} \quad (9)$$

where $\alpha$ and $\beta$ are coefficients to balance the losses. It's worth highlighting that this self-distillation loss is highly adapted to the temporal-wise distillation framework, harmoniously augmenting its efficacy using only information from the existing backbone pathway, without adding any extra feedforward computational branches. Its mechanism is structurally akin to using soft labels from the ANN, ensuring consistency across the definitions of loss.

### 3.3. Convergence of Temporal-wise Distillation

To elucidate the connection between temporal-wise distillation $\mathcal{L}_{\text{TWKD}}$ and standard distillation $\mathcal{L}_{\text{SKD}}$, we start by examining the convergence of BPTT-based SNNs' objectives. (Deng et al., 2022) points out the convergence challenges of classification objectives and suggests optimizing outputs of each timestep to avoid falling into local minima with low prediction errors but high second-order moments. The essential convergence proofs can be provided for the temporal-wise cross-entropy training objective, as in the following lemma:

**Algorithm 1** Temporal-wise Distillation Framework for Training Deep Spiking Neural Networks

---

**Require:** Pre-trained ANN model $f_{ann}$, SNN model $f_{snn}$, timesteps $T$, hyper-parameter $\alpha, \beta, \tau$, input sample $\mathbf{x}$, target label $\mathbf{y}$.

**Ensure:** Train SNN model with logits-based distillation

 1: Obtain SNN temporal outputs $\{\mathbf{z}^S(t)\}_{t \leq T} = f_{snn}(\mathbf{x})$;
 2: Obtain ANN output logits $\mathbf{z}^A = f_{ann}(\mathbf{x})$;
 3: Compute ensemble voting output $\mathbf{z}^S_{\text{ens}} = \frac{1}{T} \sum_t \mathbf{z}^S(t)$;
 4: Compute $\mathcal{L}_{\text{TWCE}} = \frac{1}{T} \sum_t \mathcal{L}_{\text{CE}} \left( \mathbf{S}(\mathbf{z}^S(t)), \mathbf{y} \right)$ in Eq. (5) by the truth target $\mathbf{y}$;
 5: Get $\mathcal{L}_{\text{TWKL}} = \frac{1}{T} \sum_t \mathcal{L}_{\text{KL}} \left( \mathbf{S} \left( \mathbf{z}^S(t)/\tau \right), \mathbf{S} \left( \mathbf{z}^A/\tau \right) \right)$ in Eq. (6) by the teacher label $\mathbf{z}^A$;
 6: Get $\mathcal{L}_{\text{TWSD}} = \frac{1}{T} \sum_t \mathcal{L}_{\text{KL}} \left( \mathbf{S} \left( \mathbf{z}^S(t)/\tau \right), \mathbf{S} \left( \mathbf{z}^S_{\text{ens}}/\tau \right) \right)$ in Eq. (8) by the ensemble label $\mathbf{z}^S_{\text{ens}}$;
 7: Obtain the final objective $\mathcal{L}_{\text{final}} = \mathcal{L}_{\text{TWCE}} + \alpha \mathcal{L}_{\text{TWKL}} + \beta \mathcal{L}_{\text{TWSD}}$ in Eq. (9);
 8: Update parameters of SNN model based on $\mathcal{L}_{\text{final}}$.

---

**Lemma 1.** $\mathcal{L}_{TWCE}$ *forms the upper bound of* $\mathcal{L}_{SCE}$, *as:*

$$
\begin{aligned}
\mathcal{L}_{\text{SCE}} &= - \sum_i y_i \log S_i \left( \mathbf{z}^S_{\text{ens}}(t) \right) \\
&\leq - \frac{1}{T} \sum_t \sum_i y_i \log S_i(\mathbf{z}^S(t)) = \mathcal{L}_{\text{TWCE}}
\end{aligned}
\tag{10}
$$

The proof is provided in Appendix A.1. The equality holds if $\mathbf{z}^S(t) = \mathbf{z}^S_{\text{ens}}$ for every $t$ in $[1, T]$. Based on Lemma 1, using $\frac{1}{T} \sum_t \mathcal{L}_{\text{CE}}(\mathbf{z}^S(t), )$ instead of $\mathcal{L}_{\text{CE}}(\mathbf{z}^S_{\text{ens}}, \mathbf{y})$ for training can be viewed as optimizing the upper bound of the overall training objective. Building on Lemma 1, we can extend our understanding to the relationship between temporal-wise distillation and standard logits-based distillation:

**Proposition 2.** $\mathcal{L}_{TWKD}$ *forms the upper bound of* $\mathcal{L}_{SKD}$, *as:*

$$
\mathcal{L}_{\text{SKD}} \leq \mathcal{L}_{\text{TWKD}}
\tag{11}
$$

The proof is provided in Appendix A.2. The proposition elucidates that just as the ground-truth CE objective is decoupled over time, the soft-label objective's decoupling can also ensure convergence of the upper bounds. Therefore, convergence with $\mathcal{L}_{\text{TWKD}}$ implies the convergence of $\mathcal{L}_{\text{SKD}}$; once $\mathcal{L}_{\text{TWKD}}$ approaches zero, the original loss function $\mathcal{L}_{\text{SKD}}$ also nears zero. Furthermore, while optimizing for the decoupling objective primarily ensures convergence only to the upper bound of $\mathcal{L}_{\text{SKD}}$, the incorporation of $\mathcal{L}_{\text{TWSD}}$ functions effectively as a regularization term. This could further tighten the inequality in Eq. (11), narrowing the gap between the optimization target $\mathcal{L}_{\text{TWKD}}$ and $\mathcal{L}_{\text{SKD}}$, which ensures that optimizing the upper bound also effectively aids the convergence of the objective $\mathcal{L}_{\text{SKD}}$.

## 3.4. Convergence Across Full-Range Timesteps

It is worth noting that temporal-wise distillation not only enhances the overall model performance but also ensures good convergence for implicitly integrated models with fewer timesteps in the ensemble. While BPTT-based SNNs training requires a predefined number of timesteps $T$ as a hyperparameter, with training targets defined on the fixed timesteps' voting outputs, this usually results in models that are tailored to specific timesteps and exhibit poor generalizability across different timesteps during inference (see Fig. 2a). In contrast, the temporal-wise distillation framework can ensure the convergence of implicit models, allowing a single trained model to handle various timestep scenarios. We refer to this capability as improving models of full-range timesteps, which greatly enhances the flexibility for model deployment.

**Proposition 3.** $\mathcal{L}_{TWKD}^{(T)}$ *defined on timesteps $T$ forms the scaled upper bound of inner* $\mathcal{L}_{SKD}^{(T_k)}$ *defined on* $T_k \leq T$, *as:*

$$
\mathcal{L}_{\text{SKD}}^{(T_k)} \leq \frac{T}{T_k} \mathcal{L}_{\text{TWKD}}^{(T)}
\tag{12}
$$

The proof is provided in Appendix A.3. It can be seen that $\mathcal{L}_{\text{TWKD}}^{(T)}$ is not only an upper bound for $\mathcal{L}_{\text{SKD}}^{(T)}$ over timesteps $T$ as Eq. (11), but also effectively reduces the upper bound of any implicit $\mathcal{L}_{\text{SKD}}^{(T_k)}$ over timesteps $T_k \leq T$ as Eq. (12); this aligns with our empirical findings where using temporal-wise distillation with timestep $T$ during the training phase results in good generalizability of timesteps during the inference stage (see Fig. 2b).

## 4. Experiments

In this section, we assess the effectiveness of the proposed method through experiments on CIFAR-10 (Krizhevsky et al., 2009), CIFAR-100 (Krizhevsky et al., 2009), ImageNet (Deng et al., 2009), and CIFAR10-DVS (Li et al., 2017). We conduct SNNs training on the Pytorch (Paszke et al., 2019) and SpikingJelly (Fang et al., 2023) platforms, employing BPTT with sigmoid-based surrogate functions (Fang et al., 2023). All experimental details are provided in Appendix B.

### 4.1. Performance Comparison on Benchmarks

We compare our proposed framework to both directly-trained methods and distillation-based methods on a variety of classification benchmarks, as shown on static datasets CIFAR-10/CIFAR-100 in Table 1, large-scale ImageNet in Table 2, and neuromorphic dynamic dataset CIFAR10-DVS in Table 3. The directly-trained methods listed in the tables are based on various adaptations of the surrogate-based BPTT training scheme, which are modifications specifically

*Table 1.* Performance comparison of top-1 accuracy (%) on CIFAR-10 and CIFAR-100 datasets, averaged over three experimental runs.

| Method | Model | Timestep | Top-1 Acc. (%) | |
| --- | --- | --- | --- | --- |
| | | | **CIFAR-10** | **CIFAR-100** |
| | | **Direct-training** | | |
| STBP-tdBN (Zheng et al., 2021) | ResNet-19 | 6 | 93.16 | - |
| | | 4 | 92.92 | - |
| | | 2 | 92.34 | - |
| Dspike (Li et al., 2021b) | ResNet-18 | 6 | 94.25 | 74.24 |
| | | 4 | 93.66 | 73.35 |
| | | 2 | 93.13 | 71.68 |
| TET (Deng et al., 2022) | ResNet-19 | 6 | 94.50 | 74.72 |
| | | 4 | 94.44 | 74.47 |
| | | 2 | 94.16 | 72.87 |
| RecDis (Guo et al., 2022b) | ResNet-19 | 6 | 95.55 | - |
| | | 4 | 95.53 | 74.10 |
| | | 2 | 93.64 | - |
| DSR (Meng et al., 2022) | ResNet-18 | 20 | 95.10 | 78.50 |
| SSF (Wang et al., 2023a) | ResNet-18 | 20 | 94.90 | 75.48 |
| SLTT (Meng et al., 2023) | ResNet-18 | 6 | 94.4 | 74.38 |
| OS (Zhu et al., 2023) | ResNet-19 | 4 | 95.20 | 77.86 |
| RateBP (Yu et al., 2024) | ResNet-18 | 6 | 95.90 | 79.02 |
| | | 4 | 95.61 | 78.26 |
| | | 2 | 94.75 | 75.97 |
| | ResNet-19 | 6 | 96.36 | 80.83 |
| | | 4 | 96.26 | 80.71 |
| | | 2 | 96.23 | 79.87 |
| | | **w/ distillation** | | |
| KDSNN (Xu et al., 2023b) | ResNet-18 | 4 | 93.41 | - |
| Joint A-SNN (Guo et al., 2023b) | ResNet-18 | 4 | 95.45 | 77.39 |
| | | 2 | 94.01 | 75.79 |
| | ResNet-34 | 4 | 96.07 | 79.76 |
| | | 2 | 95.13 | 77.11 |
| SM (Deng et al., 2023) | ResNet-18 | 4 | 94.07 | 79.49 |
| | ResNet-19 | 4 | 96.82 | 81.70 |
| SAKD (Qiu et al., 2024a) | ResNet-19 | 4 | 96.06 | 80.10 |
| BKDSNN (Xu et al., 2024) | ResNet-19 | 4 | 94.64 | 74.95 |
| TSSD (Zuo et al., 2024) | ResNet-18 | 2 | 93.37 | 73.40 |
| TKS (Dong et al., 2024) | ResNet-19 | 4 | 96.35 | 79.89 |
| EnOF (Guo et al.) | ResNet-19 | 2 | 96.19 | 82.43 |
| SuperSNN (Zhang et al.) | ResNet-19 | 6 | 95.61 | 77.45 |
| | | 2 | 95.08 | 76.49 |
| **Our** | ResNet-18 | 6 | **95.96** | **79.80** |
| | | 4 | **95.57** | **79.10** |
| | | 2 | **95.11** | **77.32** |
| | ResNet-19 | 6 | **97.00** | **82.56** |
| | | 4 | **96.97** | **82.47** |
| | | 2 | **96.65** | **81.47** |

tailored to the peculiarities of SNNs. The "w/ distillation" group in the tables includes schemes that incorporate distillation or self-distillation on top of directly-training. The results of our approach are consistently based on the hyper-parameter settings of $\alpha = 0.2, \beta = 0.5$ in Eq. (9). It should be noted, as shown in our theoretical analysis, that while our scheme can train full-range timestep implicit models simultaneously in large timestep training, achieving better performance at smaller timesteps than retraining individually, we have not used the method of extracting smaller timesteps from training at larger timesteps for a fair comparison. The models presented are obtained through consistent maximum timesteps setting, and further discussions on full-range implicit models will follow in the experimental section.

Comparing all results, it can be seen that the proposed distillation achieves comparable performance among all benchmarks for both directly-trained methods and distillation-based methods, proving that our approach can ensure effec-

*Table 2.* Performance comparison of top-1 accuracy (%) on ImageNet with single crop.

| Method | Model | Timestep | Acc. (%) |
|---|---|---|---|
| STBP-tdBN (Zheng et al., 2021) | ResNet-34 | 6 | 63.72 |
| | ResNet-50 | 6 | 64.88 |
| Dspike (Li et al., 2021b) | ResNet-34 | 6 | 68.19 |
| RecDis (Guo et al., 2022b) | ResNet-34 | 6 | 67.33 |
| TET (Deng et al., 2022) | ResNet-34 | 4 | 68.00 |
| OS (Zhu et al., 2023) | ResNet-34 | 4 | 67.54 |
| RateBP (Yu et al., 2024) | ResNet-34 | 4 | 70.01 |
| KDSNN (Xu et al., 2023b) | ResNet-34 | 4 | 67.18 |
| LaSNN (Hong et al., 2023) | ResNet-34 | 4 | 66.94 |
| SM (Deng et al., 2023) | ResNet-34 | 6 | 69.35 |
| | | 4 | 68.25 |
| SAKD (Qiu et al., 2024a) | ResNet-34 | 4 | 70.04 |
| TKS (Dong et al., 2024) | ResNet-34 | 4 | 69.60 |
| EnOF (Guo et al.) | ResNet-34 | 4 | 67.40 |
| **Our** | ResNet-34 | 4 | **71.04** |

*Table 3.* Performance comparison of top-1 accuracy (%) on CIFAR10-DVS, averaged over three experimental runs.

| Method | Model | Timestep | Acc. (%) |
|---|---|---|---|
| STBP-tdBN (Zheng et al., 2021) | ResNet-19 | 10 | 67.80 |
| Dspike (Li et al., 2021b) | ResNet-18 | 10 | 75.40 |
| RecDis (Guo et al., 2022b) | ResNet-19 | 10 | 72.42 |
| TET (Deng et al., 2022) | VGGSNN | 10 | 83.17 |
| SM (Deng et al., 2023) | ResNet-18 | 10 | 83.19 |
| SSF (Wang et al., 2023a) | VGG-11 | 20 | 78.00 |
| SLTT (Meng et al., 2023) | VGG-11 | 10 | 77.17 |
| SAKD (Qiu et al., 2024a) | VGG-11 | 4 | 81.50 |
| | ResNet-19 | 4 | 80.30 |
| **Our** | ResNet-18 | 4 | **83.50** |
| | | 10 | **86.40** |

tive convergence of the model's final ensemble and reduce the accuracy gap between SNNs and ANNs. This corresponds to our earlier conclusions. It is worth noting that based on the re-derivation of standard logits-based methods, the training overhead of our proposed framework is consistent with that of standard logits-based distillation. Our framework does not introduce any additional computational paths, merely altering the definition location of the loss. Like logits-based knowledge distillation, our approach, compared to directly-trained BPTT schemes, only adds the overhead of ANN inference to obtain teacher labels, making it the most efficient case among ANN-guided approaches.

### 4.2. Ablation Study

**Hyperparameter Settings of $\alpha$ and $\beta$.** The Table 4 reports the Top-1 accuracy under various settings of $\alpha$ and $\beta$, using the ResNet-18 model on the CIFAR100 dataset. Initially, we demonstrate that the ANN's distillation part, $\mathcal{L}_{\text{TWCE}}$, achieves a reasonable performance gain (79.26% *vs.* 79.56%), as shown in the upper part of the table. Subsequently, the lower part of the table illustrates that, with $\alpha$ fixed at 0.2, incorporating the self-distillation term $\mathcal{L}_{\text{TWSD}}$ leads to further improvements (79.56% *vs.* 79.80%). While $\mathcal{L}_{\text{TWSD}}$ is indispensable, the improvements are relatively stable around $\beta = 0.5$, which we select as the fixed setting for

*Table 4.* Performance comparison on hyperparameters $\alpha, \beta$ settings using ResNet-18 with $T = 6$ on the CIFAR100 dataset.

| $\beta = 0.0$ | $\alpha = 0.1$ | 0.2 | 0.3 | 0.5 | 0.8 |
|---|---|---|---|---|---|
| Top-1 (%) | 79.31 | **79.56** | 79.44 | 79.48 | 79.31 |

| $\alpha = 0.2$ | $\beta = 0.1$ | 0.2 | 0.3 | 0.5 | 0.8 |
|---|---|---|---|---|---|
| Top-1 (%) | 79.52 | 79.57 | 79.75 | **79.80** | 79.71 |

*Table 5.* Performance comparison on objectives combinations using ResNet-18 on the CIFAR100 dataset.

| $T$ | $\mathcal{L}_{\text{TWCE}}$ | w/ $\mathcal{L}_{\text{TWSD}}$ | w/ $\mathcal{L}_{\text{TWKL}}$ | w/ $\mathcal{L}_{\text{TWKL}}\&\mathcal{L}_{\text{TWSD}}$ |
|---|---|---|---|---|
| 4 | 78.58 | 78.94 | 79.05 | **79.10** |
| 6 | 79.26 | 79.63 | 79.56 | **79.80** |

our hyperparameters.

**Ablation Study of Training Objectives.** Experiments involving the ablation of training objectives were conducted, with three parts of labels being added sequentially to determine their effects. The results, summarized in Table 5, indicate that $\mathcal{L}_{\text{TWKL}}$ effectively enhances performance beyond $\mathcal{L}_{\text{TWCE}}$, which aligns with expectations and confirms the positive impact of soft labels distilled from the ANN teacher model. Furthermore, the addition of $\mathcal{L}_{\text{TWSD}}$ further enhances the distillation framework, demonstrating that this self-distillation setup acts as a beneficial regularization component for the framework. Overall, all objectives have contributed positively to the distillation framework and are compatible with one another.

**Comparison Study on Temporal Decoupling.** Experiments evaluating the impact of temporal decoupling were conducted using ResNet-18 on the CIFAR100 dataset, with results shown in Table 6. From the results, it can be concluded that temporal decoupling of cross-entropy loss ($\mathcal{L}_{\text{SCE}}$) and Kullback-Leibler divergence ($\mathcal{L}_{\text{SKL}}$) individually enhances performance over standard logits-based distillation. Furthermore, the beneficial effects of both can be additive, with the best model performance achieved when both are decoupled, which validates the effectiveness of our distillation framework based on temporal decoupling.

*Table 6.* Performance comparison of temporal decoupling on hard targets and soft labels using ResNet-18 on the CIFAR100 dataset.

| $T$ | $\mathcal{L}_{\text{SCE}}$ | $\mathcal{L}_{\text{TWCE}}$ | $\mathcal{L}_{\text{SKL}}$ | $\mathcal{L}_{\text{TWKL}}$ | Accuracy (%) |
|---|---|---|---|---|---|
| 4 | ✓ | | ✓ | | 78.32 |
| | ✓ | | | ✓ | 78.60 |
| | | ✓ | ✓ | | 78.74 |
| | | ✓ | | ✓ | **79.05** |
| 6 | ✓ | | ✓ | | 79.07 |
| | ✓ | | | ✓ | 79.15 |
| | | ✓ | ✓ | | 79.32 |
| | | ✓ | | ✓ | **79.56** |

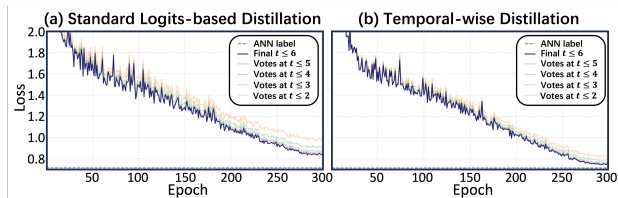

*Figure 3.* **Loss Trends**. Results of timestep ensembles during training using ResNet-18 on the CIFAR100 dataset.

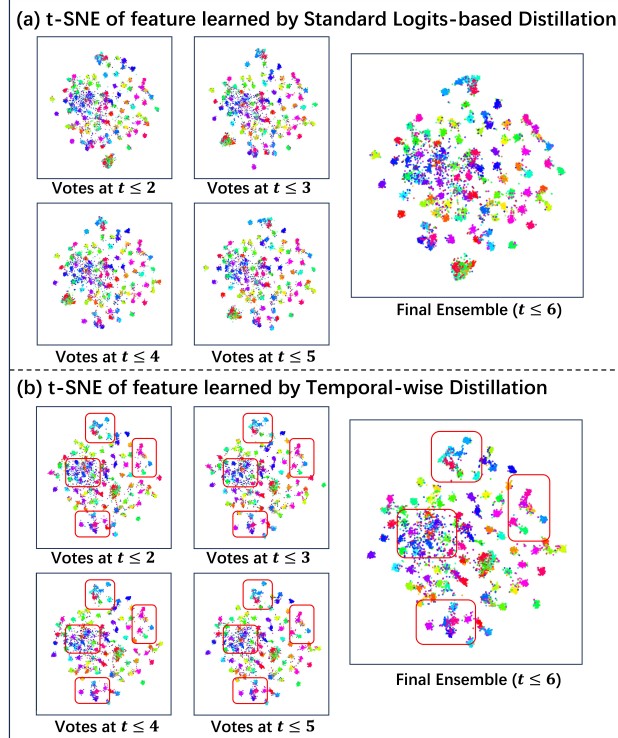

*Figure 4.* **Visual Results of t-SNE Projections**. The features are learned by (a) standard logits-based distillation and (b) the proposed temporal-wise distillation. Each subfigure progressively shows cumulative voting including more timesteps, with the final ensemble shown on the right.

### 4.3. Analysis and Discussion

**Loss Visualization.** In Fig. 3, we illustrate the convergence behavior of implicit full-range models during the training process, capturing the evolution of loss across epochs. Notably, the implementation of temporal decoupling significantly enhances the convergence of loss at each timestep. As depicted, the loss trajectories for various timestep ensembles not only improve but also exhibit a tighter and more uniform convergence compared to the standard approach. Particularly in the early phases of training with temporal decoupling, there is a notable overlap in the loss values across all timesteps. This overlapping signifies a robust synchronization in model performance, closely aligning with theoretical expectations where the loss approaches its theoretical upper bounds.

*Table 7.* Performance comparison of models trained with different timesteps (i.e. $T = 2/4/6$) and Top-1 accuracies (%) across various inference timesteps (i.e. $T = 1 \rightarrow 6$).

| Model | Training w/ | Inference Timesteps | | | | | |
|---|---|---|---|---|---|---|---|
| | | $T = 1$ | 2 | 3 | 4 | 5 | 6 |
| ResNet-18 (logits-KD) | $T = 2$ | **73.58** | **77.02** | 77.31 | 77.63 | 77.80 | 77.80 |
| | $T = 4$ | 72.44 | 76.50 | **77.57** | **78.32** | 78.47 | 78.50 |
| | $T = 6$ | 71.08 | 76.25 | 77.52 | 78.25 | **78.63** | **79.07** |
| ResNet-18 (ours) | $T = 2$ | 74.19 | 77.32 | 77.65 | 77.95 | 78.13 | 78.14 |
| | $T = 4$ | 75.08 | 77.76 | 78.40 | 79.10 | 79.21 | 79.36 |
| | $T = 6$ | **75.09** | **77.80** | **78.70** | **79.32** | **79.60** | **79.80** |
| ResNet-19 (ours) | $T = 2$ | 79.37 | 81.47 | 81.67 | 82.01 | 82.08 | 82.36 |
| | $T = 4$ | 79.40 | 81.58 | 82.14 | 82.47 | 82.39 | 82.49 |
| | $T = 6$ | **79.87** | **81.72** | **82.29** | **82.50** | **82.55** | **82.56** |

**Cluster Visualization.** As shown in Fig. 4, we present t-SNE visualizations that illustrate the clustering outcomes of two distillation strategies. The visual evidence strongly suggests that temporal-wise distillation, as depicted in Fig. 4b, results in significantly better clustering compared to the standard method shown in Fig. 4a. This enhanced clustering indicates a superior training effect on SNNs through temporal-wise distillation, consistent with outcomes from other experiments. Analyzing from the perspective of temporal ensembles, it is observed that prior to implementing temporal decoupling training (Fig. 4a), the final outcomes under different ensembles exhibit distinct separations. In the case of the proposed temporal-wise distillation in Fig. 4b, the clustering effects of the submodels exhibit a high degree of similarity. For example, the upper, lower, central-left and central-right clusters in each subplot of Fig. 4b (marked in red boxes) all display a similar pattern across the five sub-figures. This largely echoes our analysis in Section 3.4, where the loss of each submodel is essentially embedded within a larger submodel framework, resulting in a uniform convergence effect in their clustering outcomes. This uniformity signifies that the internal implicit models are converging towards the features seen in the final ensemble, corresponding to the critical role of the self-distillation component. Furthermore, this observation also explains why embedded implicit models with reduced timesteps $T_k$ perform better than those retrained at the maximum timestep $T_k$.

**Analysis of Full-Range Performance**. The results shown in Table 7 offer comparisons of models trained at different timesteps (i.e. $T = 2/4/6$) and corresponding accuracies across various inference timesteps (i.e. $T = 1 \rightarrow 6$). Horizontally, it demonstrates how inference accuracy changes when adjusting timesteps after model training. Under standard logits-KD, different models excel within specific inference timestep ranges: $T = 2$ model performs best at timesteps 1–2, $T = 4$ at times 3–4, and $T = 6$ at time 5–6. In contrast, ours consistently achieves optimal performance using the model trained at the maximum timestep ($T = 6$), clearly demonstrating the effectiveness of our method in training robust SNNs that maintain accuracy across varying inference timesteps. Vertically, the results of our proposed

method show a consistent improvement in performance as the number of training timesteps increases. The improvement across full-range inference timesteps suggests that training with a higher timestep not only enhances model accuracy but also provides a more robust generalization across varying inference lengths. In practical terms, this allows for the deployment of a single model trained at $T = 6$ to effectively replace models trained with fewer timesteps ($T = 2$ or $4$). In other words, one can utilize a fixed-parameter model to achieve comprehensive coverage across the full range of inference timesteps, significantly alleviating the stringent constraints on inference steps typically required at deployment. The flexibility in deployment is particularly advantageous, offering a streamlined approach to model utilization without sacrificing performance.

**Practical Significance of Temporal Robustness.** We conclude that ensuring the robustness of SNNs models at different inference timesteps can provide the following two technical advantages:

- **Horizontal view from Table 7.** Taking the model trained with $T = 6$ as an example, it shows stable performance across the inference window from $T = 1$ to $6$. Once deployed, the model does not require additional considerations for adaptation and switching across different inference states. This allows us to practically balance inference costs and performance directly, providing a viable model approach for scenarios that require real-time control of inference costs based on computational resources.

- **Vertical view from Table 7.** For the model trained with $T = 4$, we can invest in greater training costs to distill the $T = 6$ model, and use the submodel at $T = 4$ (essentially the same model) to achieve better performance. This offers an effective and feasible way to enhance performance by leveraging surplus training resources, providing a viable technical solution for scenarios where large training resources are available and performance enhancement is a critical issue.

We provide a more detailed discussion in Appendix C on how our proposed method can reduce the actual inference overhead of SNNs in real-world scenarios through the SEENN (Li et al., 2023) framework.

## 5. Conclusion

Leveraging the unique spatio-temporal dynamics inherent to SNNs, this work incorporates the methodology of temporal decoupling into the SNNs logits-based distillation framework. We address the deployment considerations of SNNs that typically require retraining models for different inference timesteps and provide both theoretical analysis and empirical experiments to demonstrate that our framework offers an effective solution to this issue. Experiments on standard benchmarks confirm our superior performance among distillation-based methods. By adopting temporal decoupling, our framework ensures robust model convergence and generalization across full-range timesteps. We hope this can pave the way for future developments in SNNs deployment and applications.

## Acknowledgments

This work was supported by the National Natural Science Foundation of China (Grant No. 62304203), the Natural Science Foundation of Zhejiang Province, China (Grant No. LQ22F010011), and the ZJU-YST joint research center for fundamental science.

## Impact Statement

This paper presents work whose goal is to advance the field of Machine Learning. There are many potential societal consequences of our work, none which we feel must be specifically highlighted here.

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

# A. Proof of Theorems

## A.1. Proof of Lamma 1

**Lemma 1.** $\mathcal{L}_{TWCE}$ forms the upper bound of $\mathcal{L}_{SCE}$, as:

$$\mathcal{L}_{\text{SCE}} = -\sum_i y_i \log S_i \left(\mathbf{z}_{\text{ens}}^S(t)\right) \le -\frac{1}{T} \sum_t \sum_i y_i \log S_i(\mathbf{z}^S(t)) = \mathcal{L}_{\text{TWCE}} \tag{13}$$

*Proof: Given the convex nature of the function $\log\left(\sum_{j=1}^n e^{z_j}\right)$, we know that $-\log(S_i(\mathbf{z})) = \log\left(\sum_{j=1}^n e^{z_j}\right) - z_i$ is a convex function. Here, the Hessian matrix $\mathbf{H}$ of this function is given by $\mathbf{H} = diag(\mathbf{p}) - \mathbf{p}\mathbf{p}^T$ where $\mathbf{p} = \mathbf{S}(\mathbf{z}) = [S_1(\mathbf{z}), S_2(\mathbf{z}), \dots]$. For any vector $\mathbf{v}$, we have $\mathbf{v}^T \mathbf{H} \mathbf{v} = \sum_k p_k v_k^2 - \left(\sum_k p_k v_k\right)^2 \ge 0$, resulting from the non-negativity of variance, thus $\mathbf{H}$ is positive semi-definite. Therefore, by the Jensen's Inequality, we have:*

$$\mathcal{L}_{TWCE} = \mathbb{E}\left[-\sum_i y_i \log S_i(\mathbf{z}^S(t))\right] \ge -\sum_i y_i \log S_i \left(\mathbb{E}\left[\mathbf{z}^S(t)\right]\right) = \mathcal{L}_{SCE}.$$

## A.2. Proof of Proposition 2

**Proposition 2.** $\mathcal{L}_{TWKD}$ forms the upper bound of $\mathcal{L}_{SKD}$, as:

$$\mathcal{L}_{\text{SKD}} \le \mathcal{L}_{\text{TWKD}} \tag{14}$$

*Proof: Given that $-\log(S_i(\mathbf{z}))$ is a convex function, any non-negative linear combination of such functions remains convex, i.e., for all coefficients $a_i \ge 0$, the function $-\sum_i a_i \log(S_i(\mathbf{z}))$ is convex. Therefore, by applying Jensen's Inequality, we obtain:*

$$\mathcal{L}_{TWKL} = \mathbb{E}\left[-\tau^2 \sum_i S_i \left(\mathbf{z}^A/\tau\right) \log S_i \left(\mathbf{z}^S(t)/\tau\right)\right]$$

$$\ge -\tau^2 \sum_i S_i \left(\mathbf{z}^A/\tau\right) \log S_i \left(\mathbb{E}[\mathbf{z}^S(t)]/\tau\right) = \mathcal{L}_{SKL}$$

*Together with Lemma 1, we then derive*

$$\mathcal{L}_{SKD} \le \mathcal{L}_{TWCE} + \alpha \mathcal{L}_{TWKL} = \mathcal{L}_{TWKD}.$$

## A.3. Proof of Proposition 3

**Proposition 3.** $\mathcal{L}_{TWKD}^{(T)}$ defined on timesteps $T$ forms the scaled upper bound of inner $\mathcal{L}_{SKD}^{(T_k)}$ defined on $T_k \le T$, as:

$$\mathcal{L}_{\text{SKD}}^{(T_k)} \le \frac{T}{T_k} \mathcal{L}_{\text{TWKD}}^{(T)} \tag{15}$$

*Proof: Applying Jensen's Inequality to the segments of $\sum_{t \le T} \mathbf{z}^S(t)$, specifically separating the terms into $\left[\sum_{t \le T_k} \mathbf{z}^S(t), \mathbf{z}^S(T_k + 1), \mathbf{z}^S(T_k + 2), \dots, \mathbf{z}^S(T)\right]$, we then derive:*

$$\mathcal{L}_{SKL}^{(T)} = \mathcal{L}_{KL}\left(S\left(\mathbf{z}_{ens}^S/\tau\right), S\left(\mathbf{z}^A/\tau\right)\right)$$

$$\le \underbrace{\frac{T_k}{T} \mathcal{L}_{KL}\left(S\left(\frac{1}{T_k}\sum_{t=1}^{T_k} \mathbf{z}^S(t)/\tau\right), S\left(\mathbf{z}^A/\tau\right)\right)}_{\mathcal{L}_{SKL}^{(T_k)}}$$

$$+ \frac{T - T_k}{T^2} \sum_{t=T_k+1}^{T} \mathcal{L}_{KL}\left(S\left(\mathbf{z}^S(t)/\tau\right), S\left(\mathbf{z}^A/\tau\right)\right)$$

$$\le \frac{1}{T}\sum_t \mathcal{L}_{KL}\left(S\left(\mathbf{z}^S(t)/\tau\right), S\left(\mathbf{z}^A/\tau\right)\right) = \mathcal{L}_{TWKL}^{(T)}$$

With $\mathcal{L}_{KL} > 0$, we obtain $\frac{T_k}{T}\mathcal{L}_{SKL}^{(T_k)} \leq \mathcal{L}_{TWKL}^{(T)}$. Then, similar inequalities can be applied to CE-based target objectives. Thus, we can establish the following relationship for inner implicit models: $\frac{T_k}{T}\mathcal{L}_{SKD}^{(T_k)} \leq \mathcal{L}_{TWKD}^{(T)}$.

## B. Experimental Details

### B.1. Datasets

**CIFAR-10 and CIFAR-100.** The CIFAR datasets (Krizhevsky et al., 2009) consist of 32x32 color images distributed across different classes under the MIT license. CIFAR-10 comprises 60,000 images in 10 classes, split into 50,000 for training and 10,000 for testing. CIFAR-100 includes images across 100 classes. Both datasets are normalized to zero mean and unit variance, with image augmentation techniques AutoAugment (Cubuk et al., 2019) and Cutout (DeVries & Taylor, 2017) applied. The pixel values are directly fed into the input layer at each timestep as direct encoding (Rathi & Roy, 2021).

**ImageNet.** The ImageNet-1K dataset (Deng et al., 2009) features 1,281,167 training images and 50,000 validation images across 1,000 classes, normalized for zero mean and unit variance. Training images undergo random resized cropping to 224x224 pixels and horizontal flipping, while validation images are resized to 256x256 and then center-cropped to 224x224. The pixel values are directly fed into the input layer at each timestep as direct encoding (Rathi & Roy, 2021).

**CIFAR10-DVS.** The CIFAR10-DVS dataset (Li et al., 2017) is a neuromorphic adaptation of CIFAR-10, which contains 10,000 event-based images captured by the DVS camera, licensed under CC BY 4.0. The dataset is split into 9000 training images and 1000 testing images. Data preprocessing involves integrating events into frames (Fang et al., 2021b; 2023) and reducing the spatial resolution to 48x48 through interpolation. Additional data augmentation includes random horizontal flips and random rolls within a 5-pixel range, mirroring previous methods (Xiao et al., 2022; Meng et al., 2023).

### B.2. Training Setup

**Network Architectures.** For the CIFAR-10 and CIFAR-100 datasets, we use ResNet-18 and ResNet-19 as student SNN models (He et al., 2016a; Zheng et al., 2021; Xiao et al., 2022; Fang et al., 2023; Wang et al., 2023b), applying ResNet-34 with a Top-1 accuracy of 97.24% on CIFAR-10 and 81.90% on CIFAR-100 as the teacher ANN model. In the case of the neuromorphic CIFAR10-DVS dataset, we utilize ResNet-19 as the teacher model, which is trained on the spikes mean across the temporal dimension, with a Top-1 accuracy of 83.6% for $T = 4$ and 84.4% for $T = 10$, for ResNet-18 SNN students with the corresponding timesteps. On the ImageNet dataset, our SNN model is an adapted ResNet-34 with pre-activation residual blocks (He et al., 2016b), with previous studies guiding its configuration (Xiao et al., 2022; Meng et al., 2023; Zhu et al., 2023; Yu et al., 2024). The teacher ANN model for ImageNet is a pre-trained ResNet-34 from the Timm library (Wightman et al.), which has a Top-1 accuracy of 76.32%. All SNN models incorporate the Leaky Integrate-and-Fire (LIF) neurons with a consistent membrane potential decay coefficient of 0.5, implemented in activation-based mode (Fang et al., 2023).

*Table 8.* Hyperparameters Settings.

|  | CIFAR-10 | CIFAR-100 | ImageNet | CIFAR10-DVS |
|---|---|---|---|---|
| Epoch | 300 | 300 | 100 | 300 |
| Learning rate | 0.1 | 0.1 | 0.2 | 0.2 |
| Batch size | 128 | 128 | 512 | 32 |
| Weight decay | 5e-4 | 5e-4 | 2e-5 | 5e-4 |

**Training Details.** We employ a sigmoid-based surrogate gradient method (Fang et al., 2023) to emulate the Heaviside step function with the equation $h(x, \alpha) = \frac{1}{1+e^{-\alpha x}}$ and a setting of $\alpha = 4$. The ensemble augmentation for self-distillation is implemented as (Qiu et al., 2024a). The experiments are conducted on the PyTorch (Paszke et al., 2019) and SpikingJelly (Fang et al., 2023) platforms. For CIFAR-10, CIFAR-100, and CIFAR10-DVS, we utilize a single NVIDIA GeForce RTX 3090 GPU, whereas ImageNet experiments are carried out using distributed data parallel processing across 8 NVIDIA GeForce RTX 3090 GPUs. The Stochastic Gradient Descent (SGD) optimizer (Rumelhart et al., 1986) with a momentum of 0.9 is used across all datasets, combined with a cosine annealing learning rate strategy (Loshchilov & Hutter, 2016). Detailed hyperparameters for each setup are summarized in Table 8.

# C. More Results and Discussion

## C.1. About training costs and practical effectiveness

Table 9. Results on training costs of the standard logit-based framework and the proposed framework on CIFAR-100.

| | $T = 4$ | | $T = 6$ | |
|---|---|---|---|---|
| | logits-KD | ours | logits-KD | ours |
| Time (s/batch) | 0.17367 | 0.17443 | 0.26766 | 0.26811 |
| Memory (MB) | 6333.15 | 6333.20 | 9105.12 | 9105.71 |

As a supplement to the main content, we provide measurements of training costs on CIFAR-100 in Table 9. Consistent with our discussion, the additional training overhead introduced by our method is negligible compared to the overall backbone. We conclude that our approach offers a "free lunch" in both direct training frameworks (using only $\mathcal{L}_{\text{TWCE}} + \mathcal{L}_{\text{TWSD}}$) and direct training combined with distillation frameworks (using $\mathcal{L}_{\text{TWCE}} + \mathcal{L}_{\text{TWSD}} + \mathcal{L}_{\text{TWKL}}$). We think that the practical advantages of our method are evident.

## C.2. Statistics on firing rates

Table 10. Statistical results of Firing Rates on model ResNet-18 using the CIFAR100 dataset.

| $T$ | Method | $t = 1$ | $t = 2$ | $t = 3$ | $t = 4$ | $t = 5$ | $t = 6$ | $mean$ |
|---|---|---|---|---|---|---|---|---|
| 4 | logits-KD | 0.1799 | 0.2137 | 0.2045 | 0.2091 | / | / | 0.2018 |
| | ours | 0.1819 | 0.2194 | 0.2026 | 0.2138 | / | / | 0.2044 |
| 6 | logits-KD | 0.1761 | 0.2034 | 0.2023 | 0.1966 | 0.2060 | 0.1941 | 0.1964 |
| | ours | 0.1775 | 0.2101 | 0.1937 | 0.2063 | 0.1952 | 0.1980 | 0.1980 |

We measured the average firing rate of all neurons at different time steps, as shown in Table 10. It can be observed that the proposed method exhibits firing frequencies that are essentially consistent with those of the standard logits-based method.

## C.3. Results on more spiking-based architectures

Table 11. Results on the architecture of spikingformer on the CIFAR-100 dataset.

| Arch. $[T = 4]$ | baseline | ours |
|---|---|---|
| trans-2-384 | 78.34 | 80.77 |
| trans-4-384 | 79.09 | 81.12 |

In addition to the benchmarks presented in the main content, we also verify the effectiveness of our method on Spikingformer (Zhou et al., 2023) and MS-ResNet (Hu et al., 2024a), to demonstrate its generalizability to different network architectures (Table 11 and Table 12). We note that distilling spiking transformer architectures (Qiu et al., 2025; Yao et al., 2025) requires special consideration of the choice of the teacher model ANN, as the design philosophy of spiking transformer architectures differs from traditional ANN architectures. Unlike the ResNet structure, it is not possible to find an ANN counterpart with the same structure, making this a unique consideration for transformer distillation schemes. At this point, the differences in structural design necessitate a more ingenious and novel heuristic design for the loss objective aligned with features between the ANN teacher and the SNN teacher under the feature-based distillation framework, which introduces greater design complexity in practical implementations. Therefore, we think that logits-based, end-to-end distillation offers more practical advantages in the direction of spiking transformers. As this work aptly discusses how to fully unleash the potential of logits-based distillation in exploiting the unique spatiotemporal characteristics of SNNs, it could provide a solid foundation for further exploration of logits-based distillation in spiking transformer architectures.

We also conducted experiments with VGGSNN (Deng et al., 2022) on the CIFAR10-DVS dataset, as VGGSNN is a commonly used model for this dataset. The results also allow us to better investigate the interactions among different objectives on the dynamic dataset. As shown in Table 13, $\mathcal{L}_{\text{TWSD}}$ demonstrates strong performance in this setting. As mentioned in Appendix A, since CIFAR10-DVS is a dynamic dataset, the ANN counterpart is trained on the mean input

*Table 12.* Results on the architecture of MS-ResNet on the CIFAR-100 dataset.

| Arch. $[T = 6]$ | baseline | logits-KD | ours |
|---|---|---|---|
| MS-ResNet-18 | 76.41 | 79.63 | 80.49 |

*Table 13.* Results on objectives ablation using VGGSNN on the CIFAR10-DVS dataset.

| $T$ | $\mathcal{L}_{\text{TWCE}}$ | $\mathcal{L}_{\text{TWCE}} + \mathcal{L}_{\text{TWSD}}$ | $\mathcal{L}_{\text{TWCE}} + \mathcal{L}_{\text{TWSD}} + \mathcal{L}_{\text{TWKL}}$ |
|---|---|---|---|
| 10 | 83.2 | 85.8 | 86.3 |

across the temporal dimension, which leads to lower accuracy compared to $\mathcal{L}_{\text{TWCE}} + \mathcal{L}_{\text{TWSD}}$. Nevertheless, the results indicate that the KL loss $\mathcal{L}_{\text{TWKL}}$ with soft labels still provides a positive effect on model training.

We also note some works that enhance input processing by introducing strategies at the input side (Qiu et al., 2024b; Kang et al.), which implement adaptive encoding of spike sequences during training via modules at the input layer. This could help improve the adaptability of SNNs to static inputs and better leverage SNNs' ability to process spatio-temporal information. Interestingly, under such enhanced encoding methods, intermediate features evolve over time, and spike representations embed temporal information. In this context, standard distillation frameworks may force features across time steps to align with the same ANN targets, undermining temporal diversity. Our proposed distillation framework, designed for temporal-wise decoupling, is better suited for these scenarios. We think its potential in this direction merits further investigation.

### C.4. Results toward real-world scenarios through SEENN

*Table 14.* Results of pruning inference time using the trained models under the SEENN-I framework on the CIFAR-100 dataset. We compare the standard logits-based method with our method, where SNN models are trained with $T = 4$ and $T = 6$, and then transferred to the SEENN framework for inference-time pruning. Here, CE denotes the confidence score mentioned in (Li et al., 2023), which is used to control the early-exit timing and balance the trade-off between accuracy and inference timesteps. We use "$T$ avg." to indicate the average number of timesteps during inference.

| Method | Timesteps | CS = | 0.7 | 0.8 | 0.9 | 0.99 | 0.999 |
|---|---|---|---|---|---|---|---|
| logits-KD | $T = 6$ | Acc. (%) | 73.23 | 74.65 | 77.12 | **78.75** | **79.03** |
| | | $T$ avg. | 1.139 | 1.280 | 1.606 | 2.424 | 3.076 |
| | $T = 4$ | Acc. (%) | **74.14** | **75.53** | **77.53** | 78.28 | 78.32 |
| | | $T$ avg. | 1.138 | 1.268 | 1.568 | 2.168 | 2.697 |
| ours | $T = 6$ | Acc. (%) | **76.61** | **77.58** | **79.05** | **79.75** | **79.79** |
| | | $T$ avg. | 1.165 | 1.316 | 1.690 | 2.493 | 3.188 |
| | $T = 4$ | Acc. (%) | 76.51 | 77.46 | 78.73 | 79.09 | 79.10 |
| | | $T$ avg. | 1.164 | 1.306 | 1.620 | 2.211 | 2.752 |

The SEENN project (Li et al., 2023) ingeniously designed temporal pruning, achieving a trade-off improvement between inference time and performance through early exit, and provided a dynamic adjustment scheme for inference. We replicated the SEENN-I scheme, setting it under logits-KD and our method on CIFAR-100, and compressed the inference time with the results in Table 14. First, consistent with our previous observations, logits-KD training at $T = 4$ performed better than at $T = 6$ in scenarios where the inference time was significantly reduced. We think this is because the submodel at $T = 4$ is more advantageous during moments $T = 1 \rightarrow 4$, hence at a specific time point, e.g., $t = 2$, the model trained at $T = 4$ predicts more accurately; in this case, due to the reduction in inference time, lots of models might early exit at $t = 2$, leading to a more reliable performance for the $T = 4$ model compared to the $T = 6$ model. This supports the importance of time robustness as a model property, which significantly impacts optimization when pruning inference time. The compression results of ours at $T = 4$ and $T = 6$ demonstrated the advantages brought by time robustness in actual SNN scenarios. The gains from time robustness allow us to achieve similar performance with even less reduced inference time, which can be used to further reduce the actual inference overhead of SNNs within the SEENN framework.

