# OpenReview forum: "Efficient Logit-based Knowledge Distillation of Deep Spiking Neural Networks for Full-Range Timestep Deployment"
_ICML.cc/2025/Conference — ICML 2025 poster_

### Official Review · Reviewer_BB72 · 2025-02-18

**Overall Recommendation:** 3

**Summary:**

This paper proposes Temporal-wise Logit-based Distillation, which trains the SNN timestep by timestep using the labels and the teacher output of the ANN, and uses the overall average output as a guidance signal. The authors theoretically prove the convergence of their losses and experimentally demonstrate the performance of their method on static and DVS datasets.


## update after rebuttal
Further responses from the authors demonstrate the effectiveness of their method.

**Claims And Evidence:**

The authors claim in the t-SNE visualization in Figure 4 that their method has better clustering than the standard SNN training method, but this effect is hard to perceive.

**Essential References Not Discussed:**

This paper's view of the final output of the SNN as an ensemble (in Section 3.2) is similar to [1]'s, and I suggest that the authors discuss the differences between the two.
```
[1] Rethinking Spiking Neural Networks from an Ensemble Learning Perspective. In ICLR. 2025.
```

**Experimental Designs Or Analyses:**

About the experiment:
1. Although the proposed method shows a clear performance advantage in comparative experiments, I think that **the performance advantage claimed by the authors is mainly due to the experimental setup (the performance of the baseline model) rather than to their method**. For example, using ResNet-18 at T=6, the baseline of this paper achieves an accuracy of 79.26% on CIFAR100, exceeding the performance of other methods with the same architecture. The addition of the proposed method increases the accuracy by only 0.53%, which does not indicate the performance advantage of its method. I suggest that the authors make a fair comparison with other methods under the same conditions.
2. I would like the authors to provide ablation experiments on the DVS dataset, as the use of an ANN teacher may not promote the temporal properties that are important for SNNs. In addition, I suggest that the authors add the results using VGGSNN on CIFAR10-DVS, as this is a commonly used architecture for SNN methods, and TKS [1] achieved an accuracy of 85.3% using VGGSNN on CIFAR10-DVS with 10 timesteps.
3. Could the method proposed in this paper be applied to the Transformer architecture, which might be able to improve the performance of the SOTA SNN?
4. The authors should provide ablation experiments using only $\mathcal{L}_{TWSD}$, as this is one of their most innovative points.
5. The visualization in Figure 4 does not reflect the advantages of the method proposed in this paper, and it is recommended that the authors provide more comparisons or other visualizations.
6. It is recommended that the authors report the performance of the baseline model in Table 7 to reflect the performance advantages of their method at different timesteps.

```
[1] Temporal Knowledge Sharing Enable Spiking Neural Network Learning From Past and Future. IEEE Transactions on Artificial Intelligence. 2024.
```

**Methods And Evaluation Criteria:**

The proposed method is incremental rather than innovative:
1. The temporal-wise cross-entropy (TWCE) loss proposed in this paper is the same as the TET loss (see Eq. 9 in the TET paper [1]);
2. The timestep-by-timestep distillation method is similar to the spatial self-distillation loss in TSSD [2], with the difference that this paper uses a pre-trained ANN, whereas TSSD uses the final output of the SNN to bootstrap intermediate outputs.

In addition, the authors consider the total output of the SNN as an ensemble for self-distillation, suggesting the inclusion of a discussion and comparison with [3]. Moreover, [3] also showed that their method is able to achieve better inference performance with less than the training timestep, so I suggest a fair comparison with [3].


```
[1] Temporal Efficient Training of Spiking Neural Network via Gradient Re-weighting. In ICLR. 2022.
[2] Self-Distillation Learning Based on Temporal-Spatial Consistency for Spiking Neural Networks. In Arxiv. 2024.
[3] Rethinking Spiking Neural Networks from an Ensemble Learning Perspective. In ICLR. 2025.
```

**Other Comments Or Suggestions:**

Authors should compare fairly with other methods in their experiments, not just achieve improved accuracy with a high baseline model.

In addition, there are some minor errors in the paper that need to be improved, such as an incomplete sentence in line 78 and a double citation of [1] in the references section.

```
[1] Surrogate Module Learning: Reduce the Gradient Error Accumulation in Training Spiking Neural Networks. In ICML. 2023.
```

**Other Strengths And Weaknesses:**

I think this paper's method of using the overall output of the SNN to distill the output on a timestep basis is concise and somewhat original, and I suggest that this is the main contribution of this paper. It is recommended that the authors include ablation of this loss in the experimental section to verify its validity.

**Questions For Authors:**

I have no further questions. I am willing to adjust my score if the author clearly demonstrates the innovation and effectiveness of their method.

**Relation To Broader Scientific Literature:**

I think the main contribution of this paper is similar to [1,2] with a weak core innovation.

```
[1] Temporal Efficient Training of Spiking Neural Network via Gradient Re-weighting. In ICLR. 2022.
[2] Self-Distillation Learning Based on Temporal-Spatial Consistency for Spiking Neural Networks. In Arxiv. 2024.
```

**Theoretical Claims:**

There are no obvious weaknesses in the theoretical proofs in this paper.

---

> ### Author Rebuttal · Authors · 2025-03-31
>
> # Response to Reviewer BB72
> ---
> ### __R1:__ Contributions reclaim.
> Our contributions primarily focus on a **logits-based distillation** framework tailored for SNNs, effectively leveraging spatio-temporal features **without incurring additional overhead**. Specifically, we address two critical deployment concerns: **accuracy degradation from ANNs** and **retraining costs when adjusting timestep settings**, achieving this through a lightweight, end-to-end solution validated by theoretical and empirical analysis. Furthermore, the trained SNNs demonstrate a notable advantage in terms of **robustness across inference timesteps**, as exemplified by stable performance from T=1 to T=6 (Tab. 7), enabling practical, flexible trade-offs between inference cost and accuracy. Overall, our clearly articulated methodology, novel conceptualization of temporal robustness, theoretical and empirical validation meaningfully advance the field of SNN distillation. Connections and differences with related works are further detailed in responses **R1.1**-**R1.3**, positioning our contributions within the existing literature.
> ### __R1.1:__ Relation to TET [1] and its underlying concept
> Our work relates to TET [1], which significantly advanced direct SNN training by highlighting temporal-wise decoupling of the classification objective. Inspired by this underlying concept, our paper introduces, for the first time, a clear definition and adaptation of temporal-wise decoupling into logits KD domain. Specifically, we demonstrate—both theoretically and empirically—the advantage of designing logits KD aligned with the spatio-temporal characteristics inherent to SNNs.
>
> Unlike TET, which focuses exclusively on decoupling cross-entropy loss and convergence boundary analysis, our method emphasizes applying temporal decoupling explicitly within logits KD, highlighting that performance improvements can be obtained without additional complexity (“free lunch”, **see R4 of Reviewer p1Y4**). Moreover, our work uniquely emphasizes the robustness across different inference timesteps, showcasing stable convergence for submodels (Sec. 3.4 and Tab. 7), an advantage previously unexplored by TET or related studies. Additionally, by integrating self-distillation within our temporal-decoupling framework, we fully utilize supervision signals at all timesteps without introducing extra overhead, further extending beyond TET.
> ### __R1.2:__ Relation and comparison with TSSD [2]
> We regard TSSD as representative work in self-distillation, exploring soft-label generation from temporal and spatial perspectives. Unlike TSSD, which introduces overhead through auxiliary classifiers and extra forward processes, our method achieves improvements by using existing supervision signals (teacher, ensemble, and task labels in logits-KD framework) within a fixed backbone, requiring no additional branches or training overhead.
> ### __R1.3:__ Ensemble perspective with [3]
> We recognize [3] as an excellent contemporaneous study that similarly explores submodels within SNNs from ensemble perspectives. While [3] constructs KL between adjacent timesteps (t and t-1), our method builds KL between the overall ensemble output and each individual time t. We will include discussion in the updated version, enabling readers to more comprehensively grasp ensemble learning in SNNs. We plan a detailed comparison once [3] becomes publicly available.
> ### __R2:__ About experiment setup.
> In setting up, our work has maintained strict alignment with recent SNN studies [OTTT, SLTT, ASGL, rateBP]. We replicated logits KD, TKS on C100 setup:
> ||base|logits-KD|TKS|ours|
> |-|-|-|-|-|
> |R18,C100,T=6|78.62|79.07|79.15|79.80|
>
> On Imagenet and C10-DVS (Tab. 2 & 3), base models do not exhibit special advantage. The result on large-scale __Imagenet__ is indicative of performance.
> ### __R3:__ Ablation study on CIFAR10-DVS
> ||$L_{TWCE}$|$L_{TWCE}+L_{TWSD}$|$L_{TWCE}+L_{TWSD}+L_{TWKD}$|
> |-|-|-|-|
> |vggssn,C10DVS,T=10|83.2|85.8|86.3|
> ### __R4:__ Results on spikingformer [4]
> |Arch|Spikingformer|+ours|
> |-|-|-|
> |trans-2-384,C100,T=4|78.34|80.77|
> |trans-4-384,C100,T=4|79.09|81.12|
> ### __R5:__ Ablation on TWSD
> ||$L_{SCE}$|$L_{SCE}+L_{TWSD}$|$L_{TWCE}$| $L_{TWCE}+L_{TWSD}$ |
> |-|-|-|-|-|
> |R18,C100,T=4|77.95|78.50|78.58|78.94|
> |R18,C100,T=6|78.62|79.15|79.26|79.63|
> ### __R6:__ Relation of Fig. 4 to the main claim and Tab. 7 baseline results
> Thank you for suggestions. Please refer to **R1 of Reviewer p1Y4**.
> ### __R7:__ Regarding reproducibility and minor errors
> Following modifications will ensure the code runs correctly:
> 1. In`util.py`, change`args.arch().lower()`to `args.stu_arch.lower()`within`get_model_name`function.
> 2. Delete`init_model(model)`in`bptt_model_setting`function.
> 3. In run command, replace`--epoch`with`--num_epoch`.
>
> The appendix provides necessary details. We plan to open-source complete codes and ensure reproducibility in future releases. Minor errors will be fixed, and we appreciate your feedback.

---

> > ### Comment · Reviewer_BB72 · 2025-04-05
> >
> > Thanks to the authors' response, additional experiments confirm the validity of the proposed method on the Transformer architecture as well as the TWSD loss.
> >
> > However, I still consider the rest of the losses, except for the TWSD loss, to be incremental, especially since I note that a paper [1] accepted by CVPR 2025 similarly proposes distillation by time step, which weakens the innovation of this paper. I also noticed that the results of using all losses in the ablation studies in response to R5 (78.94 and 79.63) are not consistent with the highest accuracy reported in the paper, and I hope the authors will respond to this in the next round.
> >
> > Given the incremental novelty and confusing experimental results, I don't think this paper currently meets acceptable standards for ICML. However, the experimental results provided by the authors demonstrate the effectiveness of their method, so I look forward to further responses from the authors.
> >
> > [1] Temporal Separation with Entropy Regularization for Knowledge Distillation in Spiking Neural Networks. CVPR. 2025.

---

> > > ### Author Response · Authors · 2025-04-06
> > >
> > > Thank you for your comments. We would like to further discuss our work with you.
> > >
> > > Firstly, sorry for not making it clear, the results of 78.94 and 79.63 in R5 are accurate and are based on TWCE and TWSD, excluding TWKL; the highest accuracy in our paper is achieved with all losses, which include TWCE, TWKL, and TWSD. Here, TWCE+TWSD can be considered as an ablation result of TWKL.
> > >
> > > Regarding novelty, we believe our contribution goes beyond simple modifications of the loss function; more importantly, it involves a deep analysis, exploration, and insight into the concept of time-decoupled thinking in logits-based distillation. Both theoretical analysis and empirical validations are provided to illustrate the alignment of the technical solution with its motivation. Furthermore, we delve deeper into the benefits of time robustness brought by the concept of time-decoupled, clearly demonstrating these through analysis and empirical results. To our knowledge, this discovery about the benefits of time robustness from the time-decoupled idea is first explored in our work. Overall, beyond mere performance improvement, we think the clear methodology, the new technical definition of temporal robustness in distilled SNNs explored in deployment scenarios, relevant theoretical analysis, and corresponding empirical validations undeniably contribute to the advancement of the SNN distillation field.
> > >
> > > Regarding the work [1]. [1] is a commendable concurrent work that also focuses on and demonstrates the effectiveness of time-decoupled KL loss in a distillation framework. In considering the concept of time-decoupled, our work differs from [1] in that: 1. We start with the idea of decoupling SCE into TWCE (proposed in TET), considering that CE loss is a consistency form of KL loss (where CE can be seen as a special form of KL divergence between logits and hard labels), and unify this under a comprehensive logits-based loss framework. We provide in-depth insights into the implementation of time-decoupled in logits-based knowledge distillation from perspectives of loss convergence and submodel ensemble, and offer theoretical analysis of its effectiveness. 2. We delve deeper into the benefits of time robustness brought by the concept of time-decoupled, clearly demonstrating these through both theoretical analysis and empirical validation. To our knowledge, this discovery about the benefits of time robustness from the time-decoupled idea is first explored and deeply analyzed in our work. 3. As mentioned, within this time-decoupled logits framework, we further propose TWSD, a highly compatible, cost-free self-distillation target, which pushes the concept of time-decoupled logits distillation to its limits.
> > >
> > > We appreciate you pointing out [1], which was accepted at CVPR 2025, published after we submitted this paper to ICML. We will include a discussion on its relation and differences to provide a more comprehensive literature context, and we appreciate your feedback.

---

### Official Review · Reviewer_zowF · 2025-03-06

**Overall Recommendation:** 4

**Summary:**

This paper proposes a novel distillation framework for deep SNNs that optimizes performance across full-range timesteps without
specific retraining, enhancing both efficacy and deployment adaptability. We provide both theoretical analysis and empirical validations to illustrate that training guarantees the convergence of all implicit models across full-range timesteps.

**Claims And Evidence:**

Yes

**Essential References Not Discussed:**

No discussion of recent paper [1] using distillation in SNNs.

No discussion of recent paper [2] in learning methods for SNNs.

[1] Quantized Spike-driven Transformer. ICLR 2025.

[2] Scaling Spike-driven Transformer with Efficient Spike Firing Approximation Training. IEE T-PAMI 2025.

**Experimental Designs Or Analyses:**

Yes

**Methods And Evaluation Criteria:**

Yes

**Other Comments Or Suggestions:**

See above.

**Other Strengths And Weaknesses:**

S:

a novel knowledge distillation framework is proposed. This framework optimizes the performance of SNNs over the entire time step range by means of time decoupling, without the need to retrain for specific time steps, significantly improving the deployment adaptability of SNNs.

well-written


W:

Is the proposed method still in MS-ResNet [1]?

No discussion of recent paper [2] using distillation in SNNs.

No discussion of recent paper [3] in learning methods for SNNs.

[1] Quantized Spike-driven Transformer. ICLR 2025.

[2] Scaling Spike-driven Transformer with Efficient Spike Firing Approximation Training. IEE T-PAMI 2025.

[3] Gated attention coding for training high-performance and efficient spiking neural networks. AAAI 2024.

**Questions For Authors:**

Is the proposed method still in MS-ResNet [1]?

Discuss the recent paper [2] using distillation in SNNs.

Discuss the recent paper [3] in learning methods for SNNs.

[1] Quantized Spike-driven Transformer. ICLR 2025.

[2] Scaling Spike-driven Transformer with Efficient Spike Firing Approximation Training. IEE T-PAMI 2025.

[3] Gated attention coding for training high-performance and efficient spiking neural networks. AAAI 2024.

**Relation To Broader Scientific Literature:**

N/A

**Theoretical Claims:**

Yes

---

> ### Author Rebuttal · Authors · 2025-03-31
>
> # Response to Reviewer zowF
> ---
> ### __R1:__ Is the proposed method still in MS-ResNet [1]?
>
>
> Following the experimental setup in [3], our frameword can also support the MS-ResNet [4] structure as expected, with the experimental results as follows:
>
> |CIFAR100,T=6 | baseline | logits-KD | ours |
> |-|-|-|-|
> |MS-ResNet-18|76.41|79.63|80.49|
>
> ### __R2:__ Discuss the recent paper [2] using distillation in SNNs.
>
> Thank you for your comment. We acknowledge that [2] is an excellent piece of work and represents a significant advancement in the study of spiking transformer architectures. We note that distilling spiking transformer architectures requires special consideration of the choice of the teacher model ANN, as the design philosophy of spiking transformer architectures differs from traditional ANN architectures. Unlike the ResNet structure, it is not possible to find an ANN counterpart with the same structure, making this a unique consideration for transformer distillation schemes. At this point, the differences in structural design necessitate a more ingenious and novel heuristic design for the loss objective aligned with features between the ANN teacher and the SNN teacher under the feature-based distillation framework, which introduces greater design complexity in practical implementations. Therefore, we believe that logits-based, end-to-end distillation offers more practical advantages in the direction of spiking transformers. Our work aptly discusses how to fully unleash the potential of logits-based distillation in exploiting the unique spatiotemporal characteristics of SNNs, thereby laying a solid foundation for further exploration of logits-based distillation in spiking transformer architectures.
>
> We will update our paper with the above discussion to strengthen our positioning in the literature. Thank you for your suggestion.
>
> ### __R3:__ Discuss the recent paper [3] in learning methods for SNNs.
>
> The work in [3] is quite fascinating. We understand it as implementing adaptive encoding of spike sequence inputs during the network training process by adding an attention module at the input layer, which significantly enhances the adaptability of SNN architectures to static inputs and better utilizes the SNN’s capability to process spatiotemporal information. Interestingly, under this encoding method, the characteristics of the intermediate features at different times change with time, and the spike representations carry temporal dimension information. In this scenario, the standard distillation framework would cause the features at different times to fit towards the same ANN features, thus diluting the desired diversity in the temporal dimension. The prposed distillation framework considered in our paper, which is designed for temporal-wise decoupling, is better suited for such scenarios. We think the great potential of the proposed distillation framework in this direction is worth further exploration.
>
> We will include the above discussion in the updated version. Thank you for your suggestion.
>
> ---
>
> ## Reference:
> [1] Quantized Spike-driven Transformer. ICLR 2025.
>
> [2] Scaling Spike-driven Transformer with Efficient Spike Firing Approximation Training. IEEE TPAMI 2025.
>
> [3] Gated attention coding for training high-performance and efficient spiking neural networks. AAAI 2024.
>
> [4] Advancing Spiking Neural Networks towards Deep Residual Learning. IEEE TNNLS 2024.

---

> > ### Comment · Reviewer_zowF · 2025-04-02
> >
> > I appreciate your detailed explanation. I decided to raise my score from 3 to 4.

---

### Official Review · Reviewer_p1Y4 · 2025-03-10

**Overall Recommendation:** 4

**Summary:**

This paper proposes a Temporal-wise logit-based distillation method (TWLD), which consists with three different loss (classification loss, distilation loss, and self-distillation loss) for SNN output at each time step. This new method significantly improves the performance of SNNs and their robustness to time step change.

**Claims And Evidence:**

Yes, the authors provide theoretical proof and sufficient experimental verification.

**Essential References Not Discussed:**

NA

**Experimental Designs Or Analyses:**

Most of the experiments provided by the this paper are reasonable and effective. However, I think the t-SNE experiment (Figure.4) is unnecessary as it does not demonstrate the effectiveness of the new method compared to Standard Logits-based Distillation.

**Methods And Evaluation Criteria:**

Yes, the proposed method has improved the performance upper bound of SNN, which may make sense for the research and application of SNN.

**Other Comments Or Suggestions:**

Figure 1 should provide the comparison between the proposed method and the original distillation learning, rather than direct training.

**Other Strengths And Weaknesses:**

Strength:
1. The algorithm provided by the author is simple and effective, achieving SOTA  performance on ImageNet.
2. The author provided enough theoretical proof and relevant experiments to demonstrate the effectiveness of the algorithm.

Weakness:
1. The author needs to explain in detail the practical significance of achieving a more robust SNN with time step changing.
2. The author did not provide whether their method affects the important SNN factors such as firing frequency.
3. The results presented in Table 6 show that compared to the original distillation algorithm, this paper only improved by 0.73%. The author needs to provide more detailed comparisons on the training cost of the algorithm.

**Questions For Authors:**

NA

**Relation To Broader Scientific Literature:**

NA

**Theoretical Claims:**

Yes, the authors provide theoretical proofs of the effectiveness of the corresponding claims.

---

> ### Author Rebuttal · Authors · 2025-03-31
>
> # Response to Reviewer p1Y4
> ---
> ### __R1:__ Relation of Fig. 4 to the main claim and Tab. 7 baseline results
>
> Thank you for your suggestion. We use Fig. 4 to show the distinction from the standard logits-based distillation. In the case of the proposed temporal-wise distillation in Fig. 4b, the clustering effects of the submodels exhibit a high degree of similarity. For example, the upper, lower, and central-left clusters in each subplot of Fig. 4b (which will be marked in the updated images) all display a similar pattern across the five sub-figures. This largely echoes our analysis in Section 3.4, where the loss of each submodel is essentially embedded within a larger submodel framework, resulting in a uniform convergence effect in their clustering outcomes. In the updated version, we will specifically highlight the observed clustering phenomena in Fig. 4b to better complement the discussion mentioned above.
>
> Additionally, we think the following results can more intuitively demonstrate the effectiveness of the proposed method compared to standard logits-based distillation paired with Fig 4:
>
> **Table X1. Comparison of inference performance based on models trained over different timesteps. The experiments were conducted using ResNet-18 on the CIFAR-100 dataset.**
>
> |Method|Timesteps|T=1|T=2|T=3|T=4|T=5|T=6|
> |-|-|-|-|-|-|-|-|
> |standard logits-KD|T=2|**73.58**|**77.02**|77.31|77.63|77.80|77.80|
> ||T=4|72.44|76.50|**77.57**|**78.32**|78.47|78.50|
> ||T=6|71.08|76.25|77.52|78.25|**78.63**|**79.07**|
> |ours|T=2|74.19|77.32|77.65|77.95|78.13|78.14|
> ||T=4|75.08|77.76|78.40|79.10|79.21|79.36|
> ||T=6|**75.09**|**77.80**|**78.70**|**79.32**| **79.60**|**79.80**|
>
> The table above shows inference results, supplementing the baseline results shown in Tab. 7. Horizontally, it demonstrates how inference accuracy changes when adjusting timesteps after model training. Under standard logits-KD, different models excel within specific inference timestep ranges: T=2 model performs best at timesteps 1–2, T=4 at times 3–4, and T=6 at time 5–6. In contrast, ours consistently achieves optimal performance using the model trained at the maximum timestep (T=6), clearly demonstrating the effectiveness of our method in training robust SNNs that maintain accuracy across varying inference timesteps.
>
> Thank you for your suggestion; we will include these results and analysis in the updated version.
>
> ### __R2:__ Practical significance of temporal robustness
>
> Ensuring the robustness of SNNs models at different inference timesteps can provide the following two technical advantages:
> 1. **Horizontal view from Table X1.** Taking the model trained with T=6 as an example, it shows stable performance across the inference window from T=1 to 6. Once deployed, the model does not require additional considerations for adaptation and switching across different inference states. This allows us to practically balance inference costs and performance directly, providing a viable model approach for scenarios that require real-time control of inference costs based on computational resources.
> 2. **Vertical View from Table X1**. For the model trained with T=4, we can invest in greater training costs to distill the T=6 model, and use the submodel at T=4 (essentially the same model) to achieve better performance. This offers an effective and feasible way to enhance performance by leveraging surplus training resources, providing a viable technical solution for scenarios where large training resources are available and performance enhancement is a critical issue.
>
> ### __R3:__ Results of firing rates
>
> |R18,C100|t=1|t=2|t=3|t=4|t=5|t=6|avg|
> |-|-|-|-|-|-|-|-|
> |logits-KD [T=4]|0.1799|0.2137|0.2045|0.2091|/|/|0.2018|
> |ours [T=4]|0.1819|0.2194|0.2026|0.2138|/|/|0.2044|
> |logits-KD [T=6]|0.1761|0.2034|0.2023|0.1966|0.2060|0.1941| 0.1964|
> |ours [T=6]|0.1775|0.2101|0.1937|0.2063|0.1952|0.1980|0.1980|
>
> ### __R4:__ About training costs and practical effectiveness
>
> Thank you for your suggestion. We have added measurements of training costs:
>
> ||logits-KD [T=4] | ours [T=4] | logits-KD [T=6] | ours [T=6] |
> |-|-|-|-|-|
> |Time (s/batch)|0.17367|0.17443|0.26766|0.26811|
> |Memory (MB)|6333.15|6333.20|9105.12|9105.71|
>
> Consistent with the discussion, the extra training overhead from ours is negligible compared to the overall backbone. We will include results in the updated version to reinforce the lightweight nature.
>
> Following your comments, we conclude that ours achieves a 0.73% performance improvement over logits-KD at virtually no extra cost and also introduces robustness across varying inference timesteps. We think advantages of ours in practical use are evident.
>
> ### __R5:__ Include logits-KD in Fig. 1
> Thank you for your suggestion. We will add the performance of logits-based KD to Fig. 1 to more clearly show effectiveness. The necessary data is already available in Table X1.

---

> > ### Comment · Reviewer_p1Y4 · 2025-04-05
> >
> > Thanks for the authors' response to my questions. I raised my score to 4, but I still think there is a slight gap before this paper can be directly accepted. I think that authors should discuss in more detail the actual SNN scenarios where their method is suitable instead of just raising performance. For example, they could discuss how the method can reduce the actual inference overhead of SNN through the SEENN method.

---

> > > ### Author Response · Authors · 2025-04-06
> > >
> > > Thank you for your suggestion. We noted that the SEENN project ingeniously designed temporal pruning, achieving a trade-off improvement between inference time and performance through early exit, and provided a dynamic adjustment scheme for inference. We replicated the SEENN-I scheme, setting it under logits-KD and our method on CIFAR-100, and compressed the inference time with the following results:
> > >
> > > | CIFAR100| CS | 0.7| 0.8   | 0.9   | 0.99  | 0.999  |
> > > |-|-|-|-|-|-|-|
> > > | logits-KD,T6  | acc         | 73.23 | 74.65 | 77.12 | **78.75** | **79.03**  |
> > > | | avg_time         | 1.139| 1.280| 1.606| 2.424| 3.076 |
> > > | logits-KD,T4  | acc         | **74.14** | **75.53** | **77.53** | 78.28 | 78.32  |
> > > |                | avg_time        | 1.138| 1.268| 1.568| 2.168 | 2.697 |
> > > | ours, T6       | acc         | **76.61** | **77.58** | **79.05** | **79.75** | **79.79**  |
> > > |                | avg_time         | 1.165| 1.316| 1.690| 2.493| 3.188 |
> > > | ours, T4       | acc         | 76.51 | 77.46 | 78.73 | 79.09 | 79.10  |
> > > |                | avg_time         | 1.164| 1.306| 1.620| 2.211| 2.752|
> > >
> > > First, consistent with our previous observations, logits-KD training at T=4 performed better than at T=6 in scenarios where the inference time was significantly reduced. We think this is because the submodel at T=4 is more advantageous during moments T=1~4, hence at a specific time point, e.g., t=2, the model trained at T=4 predicts more accurately; in this case, due to the reduction in inference time, lots of models might early exit at t=2, leading to a more reliable performance for the T=4 model compared to the T=6 model. This supports the importance of time robustness as a model property, which significantly impacts optimization when pruning inference time. The compression results of ours at T=4 and T=6 demonstrated the advantages brought by time robustness in actual SNN scenarios. The gains from time robustness allow us to achieve similar performance with even less reduced inference time, which can be used to further reduce the actual inference overhead of SNNs within the SEENN framework. Thank you for pointing out SEENN as a good example of considering actual scenarios. We will include it in our work to better demonstrate the effectiveness and potential application space of our work, and we appreciate your comments.

---

### Decision · Program_Chairs · 2025-05-01

**Decision:**

Accept (poster)

**Comment:**

This paper introduces a temporal-wise logit-based distillation (TWLD) framework for training Spiking Neural Networks (SNNs), with the goal of improving robustness across a range of inference timesteps. The method incorporates classification loss, distillation loss, and self-distillation loss applied at each timestep. The authors provide theoretical analysis on convergence and evaluate the approach on standard benchmarks, including CIFAR-10, CIFAR-100, CIFAR10-DVS, and ImageNet. The method aims to enhance deployment flexibility without requiring retraining for different timesteps.

## Strengths
- **Timestep robustness**: The proposed method addresses a practical challenge in SNN deployment by training models to maintain performance across varying inference timesteps.
- **Low overhead**: The approach introduces minimal computational and memory overhead, with results showing consistent, if modest, performance improvements over standard KD baselines.
- **Broad applicability**: Experimental results cover both static and neuromorphic datasets, and the framework is shown to generalize to architectures such as MS-ResNet and Transformer-based SNNs.
- **Rebuttal clarifications**: The authors provided detailed responses to reviewer questions, including new results, ablation studies, baseline alignment, and discussion of related work.

## Weaknesses
- **Incremental novelty**: The overall structure of the method shares similarities with prior work in temporal decoupling (e.g., TET), self-distillation (e.g., TSSD), and ensemble-based training. While the authors differentiate their method conceptually, the core components are viewed by some reviewers as extensions of existing techniques.
- **Comparative baselines**: Some reviewers noted that stronger or more consistent baselines (e.g., TKS, TSSD) could have been used in the initial submission. These were addressed in the rebuttal, but not all were included in the original manuscript.
- **Literature positioning**: Recent concurrent or closely related works, such as those accepted at CVPR and ICLR 2025, were not initially discussed. The authors acknowledged these papers and included comparisons in the response.

## Rebuttal Assessment
The rebuttal was comprehensive and addressed most reviewer concerns with additional experiments and analysis. Reviewers appreciated the clarification of methodological distinctions and the added evaluations on Transformer architectures and SEENN-based pruning scenarios. Several reviewers revised their evaluations positively following the response.